# PLICOTABTRANSFORMER: FOLDING TABULAR EMBEDDINGS INTO $M$ VECTORS

## ABSTRACT

Tabular data represents the most prevalent and extensively utilized form of structured data in various domains. Traditionally dominated by tree-based algorithms, researchers are actively exploring the application of deep neural networks on tabular data. Notably, the TabTransformer (Huang et al., 2020) and FT-transformer (Gorishniy et al., 2021) showed that feeding column embeddings of the tabular features into a transformer could learn a representation of the columns and how the embeddings interact with one another. This paper introduces PlicoTabTransformer, an enhancement of the previous methods, which is designed to learn multiple representations of the column embeddings. By incorporating a transformer with multiple learnable position embeddings and a contrastive learning loss, our method learns multiple distinct and orthogonal representations (denoted as *plico vectors*) of the column embeddings. We evaluated the PlicoTabTransformer with the pytorch-frame benchmark. Our experimental demonstrated that the PlicoTab-Transformer is overall top ranked algorithm and achieves state of the art performance in several datasets compared to other deep learning method closing the gap with tree based algorithms. Our method provides an added advantage to visualise redundancies and a potential dimensionality reduction technique.

## 1 INTRODUCTION

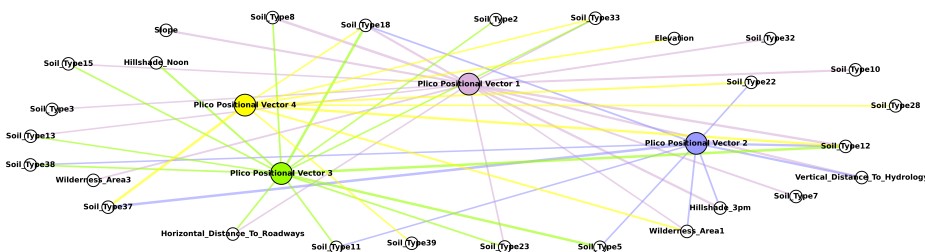

Figure 1: **Visual representation of plico vectors**: Training on the soil coverage dataset, we can assess the attention of each plico vector (centre nodes) towards each column features (outer nodes) from the soil coverage dataset. More details in Section 4.3.

Structured tabular data is among the most prevalent and extensively utilized form across various domains and industries. Sectors including healthcare (e.g., patient outcomes and treatment efficacy), financial services (market trends, assess risks), and retail (sales patterns, manage inventory) rely heavily on tabular data to enhance data-driven decision-making processes. Neural networks have great success in unstructured data including computer vision and natural language processing (NLP) beating traditional machine learning and decision tree algorithms in tasks ranging from image classification, language translation, time series forecasting, etc. However, decision tree algorithms generally still outperform perform neural networks in tabular data.

The recent works in neural networks bring down the performance gap by utilizing attention mechanisms for tabular data (Arik & Pfister, 2021; Du et al., 2021; Huang et al., 2020; Somepalli et al.,

2021; Gorishniy et al., 2022; 2021). In general, researchers have found that embedding both categorical and numerical features and feeding the embeddings into a transformer could learn a representation of the columns and how the columns interact with one another. The representation is then fed into a downstream model to predict the dataset's task.

We present the PlicoTabTransformer (*plico* is Latin for folding) that builds upon previous methods to learn multiple distinct representations of the embeded categorical and numerical tabular features. In our method, we feed the embedded input features into the transformer architecture multiple times each with a separate learnable positional embedding to encourage the transformer to attend to different columns within the dataset. Each pass of the input features through the transformer produces a representation of the embedded input features (denoted as *plico vectors*). We adapted a contrastive loss paradigm to force the plico vectors to be distinct and orthogonal to each other. The plico vectors are then fed into a downstream predictor for the dataset's task.

We evaluated our model with pytorch-frame (Hu et al., 2024), which includes state of the art decision tree algorithms and neural network models. We evaluated our method using the framework's standard benchmarking scripts, datasets, and the dataset splits. When compared to other deep learning methods, we found that PlicoTabTransformer achieved state of the art performance on a subset of the standard datasets and achieved comparable performance on the remaining datasets. Our experiments revealed that PlicoTabTransformer was the highest ranked among deep learning methods.

Our key contributions are:

- Feeding embeddings into a transformer multiple times each with different positioning embeddings to learn multiple representations of tabular data

- Creating distinct and orthogonal representations of the tabular data using contrastive learning

- PlicoTabTransformer achieves state-of-the-art performance among deep learning models

## 2 RELATED WORK

This section begins by exploring deep learning techniques applied to tabular data. We then discuss methods for learning the positional embeddings, and different contrastive loss functions.

### 2.1 TABULAR DEEP NETWORKS

Tabular deep networks are specialized for structured tabular data, using embeddings, attention mechanisms, knowledge graphs, etc. to build representations of the input data facing challenges like missing values, categorical features, and sparsity. TabNet (Arik & Pfister, 2021) introduced a novel method which uses sequential attention mechanisms to selectively process input features at each decision step. On the other hand, TabularNet (Du et al., 2021) decodes the intricate semantic structures inherent in tabular formats, going beyond traditional spatial relationships to also consider relational information between data elements. TabTransformer (Huang et al., 2020) aimed to build a strong representation for categorical features by embedding the features and feeding it into a transformer. SAINT (Somepalli et al., 2021) extends TabTransformer by integrating self-attention mechanisms not only across the features but also along the sequence of rows with a contrastive self-supervised pre-training method. By doing this, SAINT was able to capture more complex inter-feature and intra-feature relationships. FT-Transformer (Gorishniy et al., 2022) and (Gorishniy et al., 2021) further refines the transformer approach by incorporating feature tokenization, transforming categorical and numerical features into a unified representation before processing them through transformer blocks. Suggesting that input feature embeddings (for both categorical and numerical respectively) was a major contributor to improving neural network's performance. Trompt introduces prompt learning in tabular data to derive feature importances instead of focusing on the interactions among column like the regular transformer based models (Chen et al., 2023b). Authors of ExcelFormer (Chen et al., 2023a) introduce an inductive bias into the self-attention mechanism (semi-permeable attention) that selectively limits the influence of less informative features. By doing this only more informative features are permitted to propagate. In Ruiz et al. (2024), the authors use auxilary knowledge graphs describing input features to regularize multi layer perceptron. It updates each feature

embedding using a trainable message-passing function, which is optimized based on the supervised loss objective for the tabular data.

## 2.2 LEARNABLE POSITION EMBEDDINGS

Unlike recurrent neural networks, transformers do not have information on the relative or absolute position of the tokens in the sequence (Vaswani et al., 2017). Learnable positional embeddings (Gehring et al., 2017) was introduced by feeding the sequence indexes into embeddings layers to provide positional information to the neural network. Vaswani et al. (2017) showed that feeding the sequence indexes through sinusoidal functions at different frequencies could effectively inject positional information into a transformer, reducing the computational requirements. Recent works have shown the possibility of enhancing positional embeddings. Learnable sinusoidal positional encoding (LSPE) showed that feeding sinusoidal positional encoding through a feed forward network had better performance that just the sinusoidal positional encoding for document understanding tasks (Wang et al., 2022). Flow based Transformer (FLOATER) introduces a flexible positional encoding scheme that learns position information dynamically and is not restricted to the maximum length of the input (Liu et al., 2020).

## 2.3 CONSTRASTIVE LOSS

Contrastive learning aims to minimize the distance between embeddings of certain samples while maximizing the distance between embeddings of other samples. Chopra et al. (2005) trained a network to minimize the distance between image embeddings from image pairs in the same class and maximize the distance if they come from different classes. Chen et al. (2020) presented SimCLR where augmentations of images are generated and the embedding of the augmentations (positive samples) are trained to be close to the original image (anchor) and far from other images (and their augmentations). Supervised Contrastive Learning (Khosla et al., 2020) extended SimCLR incorporating label supervision, encouraging clusters of similar instances in the embedding space. StableRep (Tian et al., 2024) addresses the instability in contrastive learning by introducing methods to stabilize representation learning, ensuring robustness across diverse training scenarios.

## 3 METHODS

PlicoTabTransformer consists of three main components: columns embedding, plico vectors encoder, and downstream predictor. Let $(\boldsymbol{X}, \boldsymbol{y})$ be the feature-target pair. $\boldsymbol{X} \equiv \{\boldsymbol{X}_{cat}, \boldsymbol{X}_{cont}\}$ where $\boldsymbol{X}_{cat}$ denotes categorical features and $\boldsymbol{X}_{cont}$ continuous features. $\boldsymbol{X}$ has a total of $D$ columns, we denote $D_1$ as the number of columns of categorical features and $D_2$ as the number of columns of continuous features so $D_1 + D_2 \equiv D$. $\boldsymbol{X}_{cat} \equiv \{\boldsymbol{x}_1, \ldots \boldsymbol{x}_{D_1}\}$ where $x_i$ is a column of categorical features and $\boldsymbol{X}_{cont} \in \mathbb{R}^{D_2}$. Depending on the dataset, $\boldsymbol{y}$ could also be categorical or continuous leading to classification or regression task, respectively.

We first feed the input $\boldsymbol{X}$ into the columns embedding component to obtain embeddings for each $D$ columns. The embedded input feature is then fed through the plico vector encoder to extract $M$ distinct representations from the $D$ columns. The downstream predictor is an MLP trained for classification or regression.

The overall architecture is present in Figure 2. We describe the components in the subsequently sections.

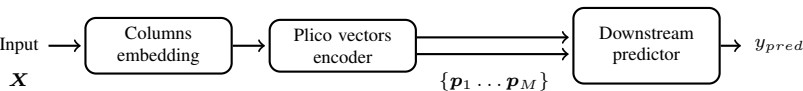

Figure 2: Overall architecture of our method.

### 3.1 COLUMNS EMBEDDING

Similar to most recent architectures for tabular data, categorical and continuous input features are separately analysed. Categorical features are tokenized and fed through an embedding layer. Continuous features are embedded through a dense layer. Similar to Trompt (Chen et al., 2023b), we fed the embedded input features through normalisation layers to ensure that categorical and continuous input features are relatively equal in magnitude.

The embedded input features are concatenated to $\boldsymbol{E} \in \mathbb{R}^{B \times D \times C}$, where $B$ is the batch size, $D$ is the number of columns in the dataset, and $C$ is the number of channels in the embedding or dense layers. The columns embedding architecture is shown in Figure 3.

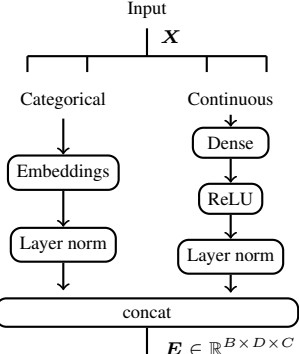

Figure 3: Architecture of the columns embedding component.

### 3.2 PLICO VECTORS ENCODER

The plico Vectors encoder takes the embedded input features $\boldsymbol{E}$ and returns $M$ plico vectors $\{\boldsymbol{p}_1, \boldsymbol{p}_2 \ldots \boldsymbol{p}_M\}$ where $\boldsymbol{p}_m \in \mathbb{R}^{B \times C}$. $M$ is a hyperparameter that could be tuned depending on the dataset.

Similar to the FT-transformer (Gorishniy et al., 2021), we model each column within $\boldsymbol{E} \equiv \{\boldsymbol{e}_1, \boldsymbol{e}_2 \ldots \boldsymbol{e}_D\}$ as a sequence and feed them through transformer layers so that each column embedding can learn to attended to the other column embeddings. Unlike the FT-transformer, we fed $\boldsymbol{E}$ through the transformer $M$ times (denoted as multi-pass) each with different learnable positional embeddings (LPE, details are described in Subsection 3.2.1). Column-wise sum was performed during each of the $M$ passes resulting in $M$ plico vectors. We included a contrastive loss to ensure that the plico vectors are distinct and orthogonal to one another (details are described in Subsection 3.2.2). Figure 4 presents the architecture.

#### 3.2.1 LEARNING POSITIONAL EMBEDDING AND MULTI-PASS TRANSFORMER

We trained $M$ separate positional embeddings to encourage the transformer to attend to different columns for each pass. We found that tokenizing the column indexes $\{1 \ldots D\}$ to an embedding layer as a mapping of the position provided the best results (Gehring et al., 2017). For each $m \in \{1 \ldots M\}$ pass, we trained positional embeddings is in the form of $\boldsymbol{PE}_m = \{\boldsymbol{pe}_{m,1}, \ldots, \boldsymbol{pe}_{m,D}\}$. The positional embedding is combined with $\boldsymbol{E}_m$ resulting in $\{\boldsymbol{e}_1 + \boldsymbol{pe}_{m,1}, \boldsymbol{e}_2 + \boldsymbol{pe}_{m,2} \ldots \boldsymbol{e}_D + \boldsymbol{pe}_{m,D}\}$ then fed into the transformer. Column-wise sum was performed on the output of the transformer to get a plico vector $\boldsymbol{p}_m \in \mathbb{R}^{B \times C}$.

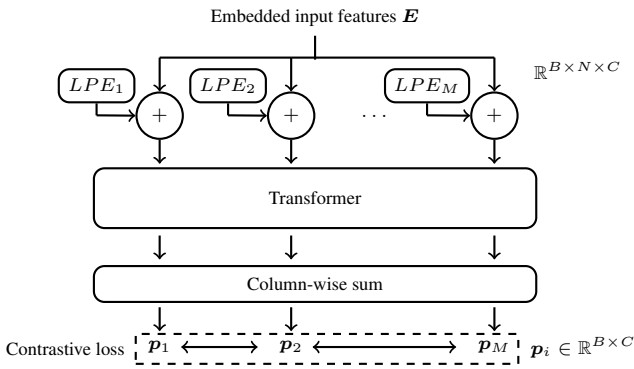

Figure 4: Architecture of the Plico Vectors encoder.

### 3.2.2 MAXIMISE DIFFERENCES BETWEEN PLICO VECTORS

To encourage the plico vectors to be distinct and orthogonal, we included a loss function on the plico vectors where vectors from the same $m^{\text{th}}$ pass are pulled together and vectors from other passes are pushed away, similar to a contrastive learning paradigm.

We adapted the self-supervised contrastive loss, as described in (Khosla et al., 2020), to our application that takes the form of:

$$\mathcal{L}_c = \sum_{i \in I} log \frac{exp(\boldsymbol{p}_i \cdot \boldsymbol{p}_{j(i)}/\tau)}{\sum_{a \in A(i)} exp(\boldsymbol{p}_i \cdot \boldsymbol{p}_a/\tau)} \tag{1}$$

where $I \equiv \{1 \ldots M\}$, $\boldsymbol{p}_i$ are plico vectors for index $i$ and $\boldsymbol{p}_{j(i)}$ are other plico vectors from $\{1 \ldots M\} \setminus \{i\}$. $\cdot$ is the dot product, $\tau$ is a temperature parameter, and $A(i) \equiv I \setminus \{i\}$.

Please note that in equation 1 of (Khosla et al., 2020), the authors designed their loss function to pull the embeddings of different augmentations (a.k.a "views") together. Our contrastive loss needs to perform the opposite (push the embeddings within the same view away from one another). Hence, we removed the negative sign in Equation 1.

We also experimented with other loss functions including the cross entropy and multi margin loss. To feed the plico vectors into the other loss functions, the plico vectors are concatenated and flattened to $\mathbb{R}^{(B \cdot M) \times C}$. The label for the loss function was generated by tiling $\{1 \ldots M\}$ $B$ times resulting in labels of size $B \cdot M$.

### 3.3 DOWNSTREAM PREDICTOR

The output plico vectors $\{\boldsymbol{p}_1, \boldsymbol{p}_2 \ldots \boldsymbol{p}_M\}$ are concatenated and fed into an MLP.

Depending if downstream task is a classification or regression problem, we used cross-entropy loss and mean squared error respectively. Combining the downstream task and plico vector contrastive loss function, we get:

$$\mathcal{L} = H + \alpha \mathcal{L}_c \tag{2}$$

where $H$ is a classification or regression loss with the labels $y$, $\mathcal{L}_c$ is the contrastive loss between plico vectors described in Section 3.2.2, and $\alpha$ is a tunable hyperparameter weighting the relative strength between $H$ and $\mathcal{L}_c$.

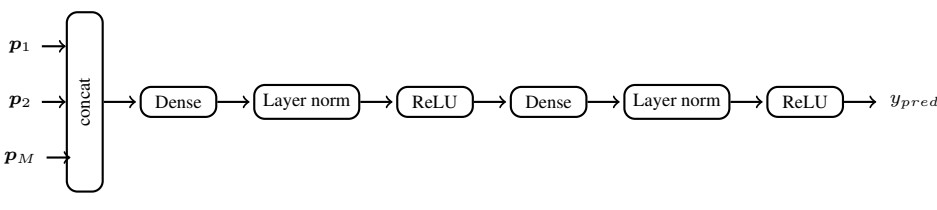

Figure 5: Downstream MLP.

## 4 EXPERIMENTS

### 4.1 SETUP

We built upon the pytorch-frame library (Hu et al., 2024) for experimentation. Pytorch-frame includes 23 binary classification datasets and 19 regression datasets collected from (Gorishniy et al., 2021; 2022; Blake, 1998). A descriptions of the datasets can be found in [1].

### 4.2 EVALUATION RESULTS

In our experiments, we directly used pytorch-frame's benchmark scripts which includes standardised splits for the data and inbuilt hyperparameter tuning (with default parameters) using optuna (Akiba et al., 2019). The hyperparameter search space for PlicoTabTransformer is presented in Table 1.

Table 1: Hyperparameters used for Plico

|  | Search space | Default DS_1 | Default DS_5 |
|---|---|---|---|
| Number of Plico vectors $M$ | [2, 4, 8, 12, 16] | 4 | 4 |
| Channels $C$ | [256, 320, 512, 768] | 320 | 768 |
| Transformer heads | [8, 16, 32, 64] | 16 | 32 |
| Transformer layers | [1, 2, 3] | 2 | 2 |
| Alpha $\alpha$ | [0.01, 0.05, 0.1] | 0.05 | 0.05 |
| Batch size | [128, 256] | 256 | 128 |
| Learning rate | [1e-4, 5e-4] | 1e-4 | 1e-4 |

We also used pytorch-frame's implementations of existing algorithm, including Trompt (Chen et al., 2023b), Excelformer (Chen et al., 2023a), and TabTransformer (Huang et al., 2020), as a comparison to our method. We focused on experimenting with the *medium* datasets for binary classification. The results for the *small* dataset are provided in Appendix A.1. DS_0 was omitted as there are out of memory errors on the benchmark. The performance of our method was compared against pytorch-frame's leaderboard[1].

Table 2 presents the classification performance. Similar to the analysis in (Chen et al., 2023b), we ranked the algorithms based on the performance, where 1 is the best and 12 is the worst performing algorithm. Furthermore, we measured each algorithm's consistency by calculating the difference between the algorithm's performance and the best performing algorithm. The ranking and difference from best algorithm is shown in Table 3.

From Table 3 we can observe that our method is the highest ranked deep learning method for binary classification. Plico also has comparable performance to ExcelFormer (Chen et al., 2023a) when using the difference from best measurement for binary classification.

---

[1]https://github.com/pyg-team/pytorch-frame/tree/master/benchmark

Table 2: Binary classification performance for *medium* datasets DS_1 to 8 (AUC - higher the better)

|  | DS_1 | DS_2 | DS_3 | DS_4 | DS_5 | DS_6 | DS_7 | DS_8 |
|---|---|---|---|---|---|---|---|---|
| XGBoost | 0.955 | 0.653 | 0.986 | 0.721 | 0.998 | 0.868 | 0.888 | 0.803 |
| CatBoost | 0.956 | 0.649 | 0.986 | 0.719 | 0.987 | 0.863 | 0.896 | 0.803 |
| LightGBM | 0.955 | 0.652 | 0.986 | 0.723 | 0.997 | 0.881 | 0.914 | 0.809 |
| Trompt | 0.95 | 0.652 | 0.982 | 0.716 | 0.966 | 0.882 | 0.883 | 0.705 |
| ResNet | 0.948 | 0.649 | 0.983 | 0.705 | 0.989 | 0.871 | 0.89 | 0.719 |
| MLP | 0.946 | 0.65 | 0.978 | 0.699 | 0.991 | 0.869 | 0.883 | 0.727 |
| FTTrans.Buc. | 0.947 | 0.649 | 0.986 | 0.651 | 0.832 | 0.866 | 0.877 | 0.688 |
| ExcelFormer | 0.948 | 0.651 | 0.982 | 0.716 | 0.995 | 0.879 | 0.883 | 0.814 |
| FTTransformer | 0.946 | 0.652 | 0.981 | 0.704 | 0.984 | 0.871 | 0.878 | 0.713 |
| TabNet | 0.945 | 0.65 | 0.977 | 0.706 | 0.993 | 0.862 | 0.889 | 0.797 |
| TabTransformer | 0.942 | 0.642 | 0.98 | 0.698 | 0.968 | 0.867 | 0.873 | 0.788 |
| Plico | 0.952 | 0.652 | 0.982 | 0.716 | 0.996 | 0.873 | 0.887 | 0.808 |

Table 3: Algorithm ranking for binary classification on *medium* datasets

|  | Binary classification | |
|---|---|---|
|  | Ranking | Diff. from best |
| XGBoost | 2.625 ± 2.233 | 0.007 ± 0.009 |
| CatBoost | 4.375 ± 3.773 | 0.008 ± 0.007 |
| LightGBM | 1.125 ± 1.166 | 0.001 ± 0.002 |
| Trompt | 5.875 ± 3.295 | 0.024 ± 0.034 |
| ResNet | 5.875 ± 2.088 | 0.022 ± 0.029 |
| MLP | 7.500 ± 1.581 | 0.023 ± 0.026 |
| FTTransformerBucket | 8.375 ± 3.462 | 0.054 ± 0.058 |
| ExcelFormer | 3.875 ± 1.965 | 0.007 ± 0.009 |
| FTTransformer | 7.000 ± 2.398 | 0.025 ± 0.031 |
| TabNet | 7.000 ± 3.000 | 0.013 ± 0.007 |
| TabTransformer | 9.375 ± 1.654 | 0.021 ± 0.011 |
| Plico | 3.000 ± 1.323 | 0.007 ± 0.008 |

## 4.3 EMBEDDING VISUALIZATIONS

To analyze the representation learned by Plico, we focus on the $M$ positional embeddings (LPE in Figure 4), which determines the contribution of the categorical and numerical features towards the plico vectors. We trained the model with the default parameters in Table 1 on the soil covertype dataset[2] (input *medium* dataset DS_5).

In the first experiment, we visualised four positional embeddings using 3D t-SNE visualizations (van der Maaten & Hinton, 2008) corresponding to $M = 4$ plico vectors. Figure 6 show distinct clustering patterns, indicating that each encoder captures unique features. While there is some overlap between clusters, suggesting shared information across encoders, the variation in cluster density and separation implies that each encoder contributes differently to the model's understanding of the data. This combination of redundancy and complementary suggests that the positional encoders collectively enhance the model's ability to represent and differentiate features effectively.

We then compared the differences between the gradients that pass through the four *LPE* (shown in Figure 4). Specifically during back-propagation, we collected the gradients of the positional encoder and represented it as a 2D matrix. The cosine similarity was then used to calculate the angle between the 2D gradients and we plot them against each other in Figure 7. The graph provides a visual representation of the angles of deviation between the gradients of four positional encoders in a model, namely pos-encoder-1, pos-encoder-2, pos-encoder-3 and pos-encoder-4. Each radar plot corresponds to one positional encoder and compares its gradient's deviation with the other three

---

[2]https://www.openml.org/d/44120

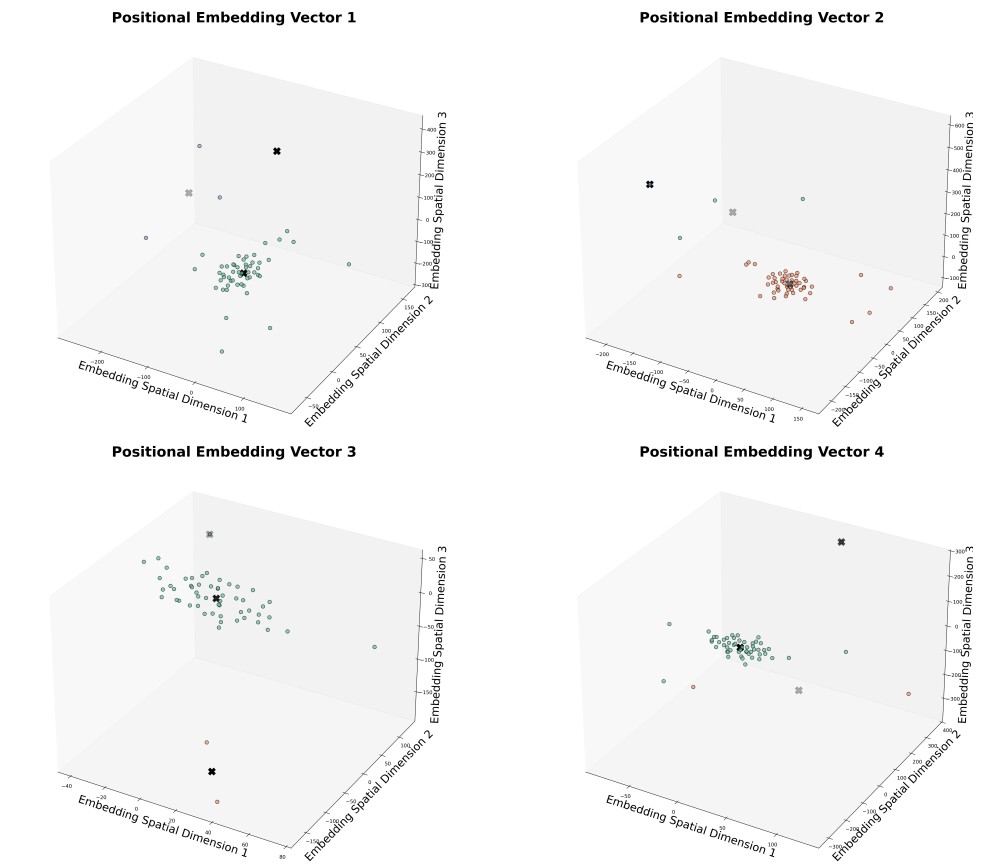

Figure 6: Visualization of four Plico position encoders trained on *medium* dataset DS_5

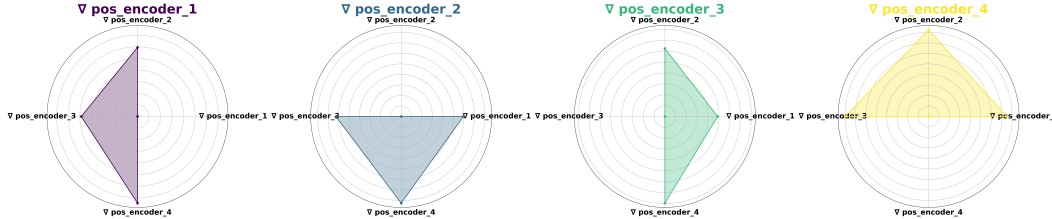

Figure 7: Angle between the gradients of Plico position vectors on *medium* dataset DS_5

encoders. A smaller angular deviation means that the encoders have more similar gradient directions (i.e., they are learning similar features), whereas larger angles indicate they are learning different positional features. For instance, in the plot for pos-encoder-1, it has a smaller deviation from pos-encoder-3, indicating these two encoders are more alike in their learning behavior, while the larger deviation from pos-encoder-4 indicates more distinct learning. The resulting angles between these gradient vectors reveal significant divergence, indicating that the positional encoders have effectively learned to capture different features or columns from the input data. This high angular separation suggests that each encoder is specializing in distinct aspects of the data, enhancing the model's ability to represent diverse features.

Finally, we wanted to provide deeper insight into how the model allocates attention across different columns of tabular inputs. The attention weights in Plico are extracted directly from the multi head attention layers within the Transformer during the forward pass. When the input passes through the transformer, weights of the attention layers are obtained. These weights represent how each position in the input attends to every other position. The weights of the final forward pass then stored for later

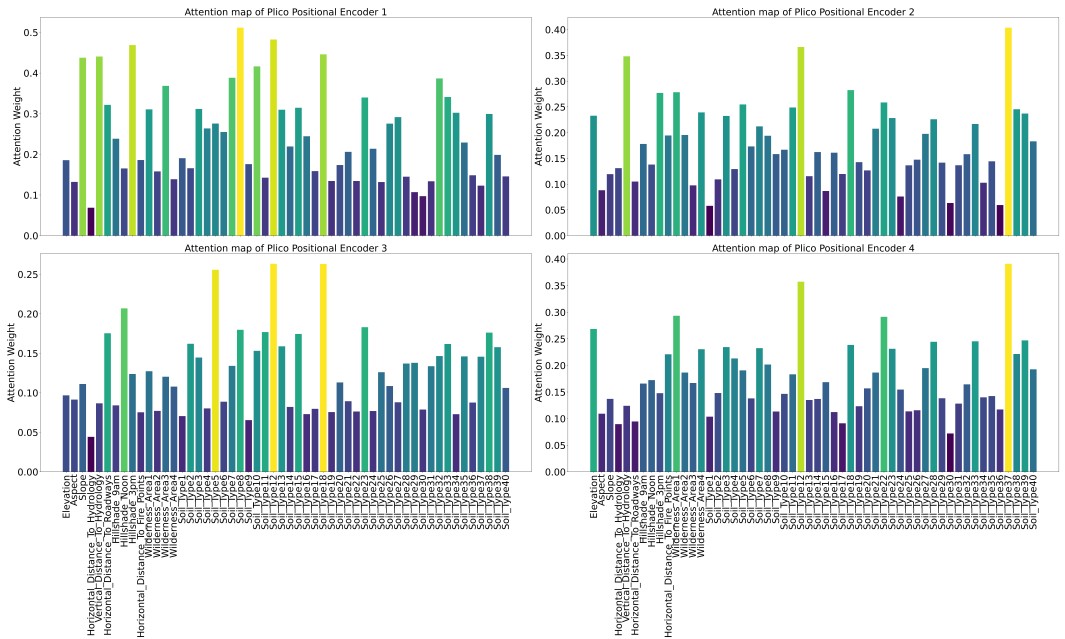

Figure 8: Attention map of Plico position encoders on *medium* dataset DS_5

visualization, allowing for analysis of the model's attention patterns across layers and heads. Figure 8 shows the heatmap visually illustrates how each of plico vectors learn distinct groups of inputs and how they receive different levels of attention, indicating that the model has learned to identify which features are most informative for making accurate predictions. The fact that the attention is distributed in a structured way across the input data suggests that Plico is capturing meaningful patterns and relationships between features, which contributes to its ability to generalize effectively across the data. By doing so Plico achieves better representation of the input data, which plays a critical role in achieving state-of-the-art results. This approach helps the model not only reduce noise from less relevant features but also effectively group and process important patterns within the data, further improving its predictive capabilities.

### 4.4 ABLATION STUDY

We conducted ablation studies to investigate the effects of different contrastive loss functions for the plico vectors and different learnable positional embeddings. For this experiment, we fixed the dataset to the KDD Census Income dataset (*medium*, binary classification - $dataset\_1$)[3] and the hyperparameters to the default values as shown in Table 1.

### 4.4.1 PLICO VECTORS CONTRASTIVE LOSSES

Table 4 shows that the adapted self-supervised contrastive loss presented in Equation 1 had the best results. In general, including a contrastive loss function on the plico vectors had improvements compared to not included a loss function. We also presented the results of the contrastive loss as is from Khosla et al. (2020), which is designed to pull the embeddings from different passes together. This loss had degradation of performance even compared to not including a loss function.

### 4.4.2 LEARNABLE POSITIONAL EMBEDDINGS

From Table 5, we can observe that using learnable positioning embeddings outperform sinusoidal positional embeddings described in (Vaswani et al., 2017). Furthermore, a standard embedding layer had better performance compared to LSPE (Wang et al., 2022). With multiple passes to the trans-

---

[3]https://archive.ics.uci.edu/dataset/117/census+income+kdd

Table 4: Performance of plico contrastive losses

|  | AUC |
| --- | --- |
| Self-supervised contrastive loss (Khosla et al., 2020) | 0.9497 |
| No loss function | 0.9514 |
| Cross entropy | 0.9513 |
| Stable Rep (Tian et al., 2024) | 0.9516 |
| Multi margin loss | 0.9515 |
| Adapted self-supervised contrastive loss (Equation 1) | 0.9518 |

former, it is clear that using learnable positional embeddings could shift attention towards different columns for each $m$ pass.

Table 5: Performance of learnable positional embeddings

|  | Learnable | AUC |
| --- | --- | --- |
| Sinusoidal (Vaswani et al., 2017) | $\times$ | 0.9511 |
| LSPE (Wang et al., 2022) | $\checkmark$ | 0.9515 |
| Embeddings (Gehring et al., 2017) | $\checkmark$ | 0.9518 |

## 5 CONCLUSION

In this paper, we introduced the PlicoTabTransformer, a novel approach that leverages multiple passes of data through a transformer model with separate learnable position embeddings to learn multiple distinct and orthogonal representations of tabular datasets. Our method demonstrated state-of-the-art performance when compared to existing deep learning techniques in a subset of the datasets and was among the top ranked deep learning algorithms.

Given the inherent diversity in structured tabular datasets, including variations in the number of columns and the nature of column data, it is evident that different algorithms have different advantages. We believe that PlicoTabTransformer is a compelling option among the available algorithms which could be used for tabular data.

To our knowledge, PlicoTabTransformer is among the first works to perform multiple passes of embeddings into transformer with multiple positional embeddings and creating distinct representations with contrastive learning. We hope that researchers could build upon this framework and apply this method to other neural network architecture and applications.

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

# A APPENDIX

## A.1 *Small* DATASET PERFORMANCE

Table 6: Binary classification *small* dataset results

| | 0 | 1 | 2 | 3 | 4 | 5 | 6 | 7 | 8 | 9 | 10 | 11 | 12 | 13 |
|---|---|---|---|---|---|---|---|---|---|---|---|---|---|---|
| XGBoost | 0.931 | 1 | 0.94 | 0.947 | 0.885 | 0.966 | 0.862 | 0.779 | 0.984 | 0.714 | 0.787 | 0.951 | 0.999 | 0.925 |
| CatBoost | 0.93 | 1 | 0.938 | 0.924 | 0.881 | 0.963 | 0.861 | 0.772 | 0.93 | 0.628 | 0.796 | 0.948 | 0.998 | 0.926 |
| LightGBM | 0.931 | 0.999 | 0.943 | 0.943 | 0.887 | 0.972 | 0.862 | 0.774 | 0.979 | 0.732 | 0.787 | 0.951 | 0.999 | 0.927 |
| Trompt | 0.919 | 1 | 0.945 | 0.942 | 0.881 | 0.964 | 0.855 | 0.778 | 0.933 | 0.686 | 0.793 | 0.952 | 1 | 0.916 |
| ResNet | 0.917 | 1 | 0.937 | 0.938 | 0.865 | 0.96 | 0.828 | 0.768 | 0.925 | 0.665 | 0.794 | 0.946 | 1 | 0.911 |
| MLP | 0.913 | 1 | 0.934 | 0.938 | 0.863 | 0.953 | 0.83 | 0.769 | 0.903 | 0.666 | 0.789 | 0.94 | 1 | 0.91 |
| FTTransformerBucket | 0.915 | 0.999 | 0.936 | 0.939 | 0.876 | 0.96 | 0.857 | 0.771 | 0.909 | 0.636 | 0.788 | 0.95 | 0.999 | 0.913 |
| ExcelFormer | 0.918 | 1 | 0.939 | 0.939 | 0.883 | 0.969 | 0.833 | 0.78 | 0.94 | 0.67 | 0.794 | 0.95 | 0.999 | 0.919 |
| FTTransformer | 0.918 | 1 | 0.94 | 0.936 | 0.874 | 0.959 | 0.828 | 0.773 | 0.909 | 0.635 | 0.79 | 0.949 | 1 | 0.912 |
| TabNet | 0.911 | 1 | 0.931 | 0.937 | 0.864 | 0.944 | 0.828 | 0.771 | 0.913 | 0.606 | 0.79 | 0.936 | 1 | 0.91 |
| TabTransformer | 0.91 | 1 | 0.928 | 0.918 | 0.829 | 0.928 | 0.816 | 0.757 | 0.885 | 0.652 | 0.78 | 0.937 | 0.996 | 0.905 |
| PlicoTabTransformer | 0.917 | 0.999 | 0.943 | 0.935 | 0.875 | 0.962 | 0.856 | 0.775 | 0.928 | 0.640 | 0.793 | 0.948 | 0.997 | 0.911 |

