# OpenReview forum: "PlicoTabTransformer: Folding Tabular Embeddings Into M Vectors"
_ICLR.cc/2025/Conference — ICLR 2025 Conference Withdrawn Submission_

### Official Review · Reviewer_WeJk · 2024-11-03

**Soundness:** 2
**Presentation:** 2
**Contribution:** 2
**Rating:** 3
**Confidence:** 4

**Summary:**

The paper introduces a new transformer-based model, PlicoTabTransformer. It incorporates learnable positional embeddings and maximizes the differences between output vectors across multiple passes of the input, aiming to learn multiple distinct representations of the column embeddings.

**Strengths:**

The methods are clearly described.

**Weaknesses:**

The authors mention, “We trained M separate positional embeddings to encourage the transformer to attend to different columns for each pass.” However, multihead attention in transformers already has the ability to capture different feature relationships. What is the advantage of training M separate positional embeddings and processing the input M times? Is there an ablation study on the hyperparameter 'M’?
• Additional baselines are needed for a comprehensive comparison. For instance, AMFormer[1] is another transformer model specifically designed for tabular data, TabR[2] represents a state-of-the-art model.
• Results on regression and multi-classification tasks are not presented in the paper. I'd also like to see more detailed performance on these tasks.
• The training cost could be significantly higher than that of a standard transformer, increasing by approximately M-fold. Could an analysis of the training cost, including metrics such as training time and parameter count, be provided?
• In the ablation study, the performance gain from the adapted self-supervised contrastive loss and learnable positional embeddings appears minimal.

[1] Arithmetic Feature Interaction Is Necessary for Deep Tabular Learning. [AAAI 2024]
[2] TabR: Tabular Deep Learning Meets Nearest Neighbors. [ICLR 2024]

**Questions:**

please refer to weakness

---

### Official Review · Reviewer_AYRU · 2024-11-04

**Soundness:** 1
**Presentation:** 2
**Contribution:** 1
**Rating:** 3
**Confidence:** 2

**Summary:**

This manuscript introduces PlicoTabTransformer, a transformer model designed for tabular data.
The method creates plico vectors by passing the data through the transformer multiple times, each with unique positional embeddings, allowing the model to capture diverse feature interactions.
Additionally, a contrastive training approach is employed to ensure these vectors remain distinct, enhancing the model’s ability to learn complex patterns effectively.

**Strengths:**

The proposed plico vector approach is novel, providing a unique way to capture diverse feature interactions by repeatedly embedding the data with distinct positional encodings.

**Weaknesses:**

The manuscript’s motivation is somewhat unclear.
Given that the multi-head attention mechanism in transformers is already designed to capture diverse feature interactions, it’s not immediately evident why additional passes with varied positional embeddings are necessary.

A notable drawback is the selective reporting of performance.
While the proposed method outperforms conventional models on _medium_ datasets, its performance on _small_ datasets is underwhelming.
Moreover, the absence of results on _large-scale_ datasets raises concerns about the model’s scalability and robustness in diverse real-world scenarios.

**Questions:**

1. Could the authors clarify the choice of learned positional embeddings over more recent alternatives, such as rotary embeddings? Rotary embeddings have demonstrated effectiveness in capturing position-invariant relationships in transformers and may offer advantages in handling diverse feature interactions. Understanding the rationale behind this choice could help clarify the design decisions in PlicoTabTransformer.
2. Could the authors provide performance evaluations on large-scale datasets, particularly in comparison to ExcelFormer? The current results focus on medium datasets, with limited insight into scalability. A comparison with ExcelFormer on large-scale datasets would offer a better perspective on PlicoTabTransformer’s robustness and practical applicability across different dataset sizes.

---

### Official Review · Reviewer_ZmMF · 2024-11-04

**Soundness:** 2
**Presentation:** 1
**Contribution:** 1
**Rating:** 1
**Confidence:** 4

**Summary:**

The paper proposes to do multiple passes of a transformer for tabular data with different feature positional embeddings for each pass. It then combines the resulting object representations to make a prediction. In addition to the default supervised loss function, the proposed method uses an additional loss to push those representation vectors apart.

The method is evaluated on a benchmark from the pytorch-frame library.

**Strengths:**

- Proposed architecture delivers some performance improvements on the chosen benchmark compared to the vanilla transformer for tabular data.

**Weaknesses:**

- The writing can be substantially improved. It is currently hard to understand the model. Some examples of unclear writing:
  - It is still unclear to me whether the weights of the transformer encoder that performs multiple passes are shared or not
  - figure 4 (what is N, is it supposed to be a number of features? It is D in your notation
  - L211 -- what does tokenizing the column indices mean?
  - Figures 6 and 7 do not provide any clarity or explanation of what's happening there
  - Why are bars in Figure 8 called attention maps? Are these some feature-importance derived from attention maps?
  - These are just a sample that make the paper hard to digest and fully understand.
- An important practical limitation if I understand the method correctly is efficiency. It is not discussed anywhere (except the discussion of OOM on DS_0 in L311).
- Strong contemporary tabular DL methods are missing (see Q below). I believe those methods, when properly tuned would outperform the PilcoTabTransformer (while being more efficient). Current results seem overly optimistic due to comparisons only to the quite outdated transformer architectures.
- No ablations or even intuition provided that explain the performance improvements. The only ablations regard contrastive loss variations and positional embedding variations, the core method components are not ablated. Ablation on one dataset in tabular ML is not very representative (due to the very high diversity of the datasets)

**Questions:**

- How does your method compare to stronger tabular DL models like nearest-neighbors based like ModernNCA [https://arxiv.org/abs/2407.03257] or TabR [https://arxiv.org/abs/2307.14338] and simple MLPs with embeddings for numerical features [https://arxiv.org/abs/2203.05556]?
- What are the dimensions of the input in figure 4? Shouldn't it be BxDxC, not BxNxC in your notation?
- Are the weights of the Transformer encoder shared?

---

### Official Review · Reviewer_WCjw · 2024-11-07

**Soundness:** 2
**Presentation:** 2
**Contribution:** 2
**Rating:** 5
**Confidence:** 4

**Summary:**

The paper aims to close the gap between tree-based methods and deep tabular models by proposing an enhanced tabular transformer, which learns multiple representations of the column embeddings.

For a dataset with customer data, PlicoTabTransformer would generate multiple representations (plico vectors) of each feature (e.g., age, income) by feeding the data through the transformer with different positional embeddings. This encourages the model to capture distinct patterns in customer behaviour that may not be apparent in a single representation.

**Strengths:**

* The proposed method extends the existing tabular transformer by proposing to embed features multiple times with different position encodings. This seems to be an important insight for learning robust representations of tabular data.
* The proposed method is simple and clear.
* The paper content is generally well-written.

**Weaknesses:**

**1. [Important] Rationales behind PlicoTabTransformer.** If I understand correctly, PlicoTabTransformer can be viewed as a tabular data augmentation method in the representation space. The augmentation here is to “permute the order of features” with different positional encodings. However, it seems a bit unclear to me how this can help tabular models perform better. As tabular data is expected to be permutation invariant, i.e., with different orders, the outputs should be the same. Or PlicoTabTransformer is doing such augmentation in another way? I would love to discuss the crux of the proposed method with the authors.

**2. [Important] Vague presentation of the results.** The authors seem to overly rely on the document of third-party repos, and thus the main content is not sufficiently self-contained. For instance:

    * in Line 289, the authors refer readers to the data descriptions in the link at the footnote. However, the link is deprecated and no useful information can be retrieved there.

    * The corresponding data preprocessing process are missing as well, like splitting strategies. This can lead to vagueness in the standard deviations reported in the paper as well.
Due to the above concerns, I remain conservative about the results reported in the paper. And I cannot provide any conclusive feedback for the results.

**3. [Important] Potentially limited scalability.** PlicoTabTransformer seems to concatenate the plico vectors directly before inputting them into the downstream predictor. I am wondering if this would lead to a soar in the dimensionality for downstream predictors, which could lead to the infeasibility of downstream predictors.

**4. Lack of hyperparameter analysis.** Only a narrow hyperparameter range was explored, I am a bit confused by Table 1. Do the authors perform a grid search with Optuna? Then what are the desiderata?

**5. No open-source implementation.** Code is not provided, and thus I remain conservative on the reported results in the paper.

I am happy to adjust my score according to the response from the authors.

**Questions:**

Please refer to the Weaknesses section.

---

### Note · Authors · 2024-11-21

**Comment:**

Thank you so much for the reviewers comments. It seems to us that all the reviewers agree that our method needs more experimentation and clarity. We have decided to withdraw this paper and conduct further experimentation before resubmission.

**Withdrawal Confirmation:**

I have read and agree with the venue's withdrawal policy on behalf of myself and my co-authors.